# OpenReview forum: "Post-Hoc Reasoning in Chain-of-Thought: Evidence from Pre-CoT Probes and Activation Steering"
_ICLR.cc/2026/Conference — Submitted to ICLR 2026_

### Official Review · Reviewer_aHJG · 2025-10-24

**Soundness:** 2
**Presentation:** 2
**Contribution:** 3
**Rating:** 4
**Confidence:** 4

**Summary:**

This paper studies post-hoc reasoning, i.e. the tendency of LLMs to commit to a prediction before additional chain of thought processes in tasks thought to require reasoning. In particular, it aims to uncover the latent mechanisms of post-hoc reasoning. To this end, the authors consider five language models (two from the Gemma 2 and three from the Qwen 2.5 families) across five binary QA tasks. There, through erasing or falsifiying given CoTs, the paper shows that the models exhibits post-hoc reasoning. This is furhter reinforced by linear probes of the model activations that show that the final model answer after CoT is reliably predictable across models and tasks at the last pre-CoT position.

The paper then reports steering experiments, where the steering vectors are given by the probing directions found. Through this, the binary model answers can be steered successfully causing the model to change its predictions. Finally, the CoT pathologies are classified through an LLM as a judge approach finding that confabulation and non-entailment happen equally often.

**Strengths:**

The paper looks at an interesting hypothesis of failure of CoT-style prediction in language models.

The probing experiments provide an interesting new insight, namely, that the final post- CoT prediction is reliably predictable from the last pre-CoT state.

Classifying the types of reasoning behavior in the steered predictions is an interesting approach to quantify the changes in behaviour resulting from the steering.

While there are several crucial weaknesses listed below, I'd encourage the authors to continue working on this interesting idea.

**Weaknesses:**

(W1) The models analysed are both instruction tuned versions. In the related work, Vernhoff et al., 2025 and Zhang et al., 2025 report studying ‘reasoning' or ‘thinking’ models, which refers to distilled versions of DeepSeek R1. Given that the core research question of this paper is to study CoT behavior, it is unclear why the authors did not decide to also study models trained for CoT processing. Can the results from models not trained for CoT be expected to generalize to such models?

(W2) One of the major weaknesses relates to the second key contribution. Firstly, it is not clear what the authors actually show with the steering experiments. In section 3.4, they state they want to test whether "the pre-CoT probes are merely correlated with the final answer, and do not themselves represent the final answer or causally influence it”. In Section 4.5 they state that "The results of the steering experiments are consistent with an interpretation of the pre-CoT probe as a causal representation of the pre- committed answer”. The preceeding Section 4.4 however does not make clear how the results actually imply this interpretation.

The way I understand it, the probe learns the directional difference of samples predictied “yes” versus samples predicted “no”. If we add/subtract this direction across all positions of the CoT chain, getting the answer to flip from “yes” to “no” and vice versa seems unsurprising. For the causal claim “the pre-CoT activation along the probe direction $w^(l*)$ is causing the final model answer” (i.e. post-hoc commitment) it should be shown that “in the absence of the probe direction v* from the final pre-CoT position, the model flips it’s prediction”, i.e. proving causality by showing the counterfactual to reliably result in the opposite behaviour.

To create this counterfactual, one could steer as $x_0 = x_0 - \langle w^{l*}, x_0 \rangle x_0$, i.e. eliminate the linear component of "x_0" should along the “yes direction" $w^{l*}$ (and vice versa $-w^{l*}$ for the “no direction”).

(W3) Besides, the predictions seem to flip already a lot at random without any steering (alpha=0 in Figure 4), the orthogonal baseline results in even more flips, and the most reliable steering success usually happens at alpha levels with high parse failures. This further weakens the steering results itself as well as the interesting results from Section 4.5, as only the portion of successfully steered examples is studied. The high level of hallucination there further calls into question how general classification of steered CoT behavior really is.



**Minor weaknesses, questions, and suggestions:**

– Section 4.2 references Appendix B but then proceeds with the same text as in the
Appendix. Would it make sense to leave out this refence and just reference
Appendix Figure 6?

– Some citations seem to be formatted incorrectly, e.g. line 46 “nostalgebraist
(2024)” should be “(nostalgebraist, 2024)”, same goes for line 047 “Bai et al.
(2022)”

– Is it possible that the models studied have already seen the evaluation data during
training thus making successful post-hoc prediction more likely through data
leakage?

– The link to the limitations in §5.2 does not point to the reported discussion of the
probe caveats. Those appear in §5.1 (but probably should be in §5.2 as they also
read like limitations).

– The Related Work section could use a clearer differentiation to the work presented in
the paper. What are the gaps/limitations addressed through this work compared to
existing works?

– Antropomorphisations like "To determine if the model is thinking about the final
answer” seem unscientific. Personally, I would use a more exact wording like “if the
final answer is represented”.

– Why are some values for alpha=0 missing in FIgure 5?

**Questions:**

1. Why did you not use reasoning models like Venhoff et al., 2025 (DeepSeek-R1-Distill) and Zhang et al., 2025 (R1-Distill-Llama and R1-Distill-Qwen families)? Can the results from models not trained for CoT be expected to generalize to such models?

2. Could you please clarify the hypothesis tested through the probing experiments and how the results imply this hypothesis exactly? Optimally, a mathematical definition of the variables thought to have a causal relation and how you test it.

3. Would it make sense to look into further settings/hyperparameters for the steering to obtain results with a better ratio of successfully steered samples? In the light of the questions studied, focussing on one, optimal setting for alpha seems like a good idea. Right now, the results of 4.4 and 4.5 contain ablations for different alpha values of which most result in high parse failures or low steering success, making it hard to interpret them.

---

> ### Comment · Reviewer_aHJG · 2025-11-26
>
> The authors did not provide any rebuttal answer. Therefore, keeping my score.

---

> ### Author Response · Authors · 2025-12-03
> **Response to Reviewer aHJG (1/5)**
>
> We thank the reviewer for the thoughtful review. A couple of the reviewer's concerns relate to the omission of reasoning models.  In these cases, we do not respond directly here but instead point the reviewer to the pooled response to all reviewers. We address the remaining comments below.
>
> > (W2) One of the major weaknesses relates to the second key contribution. Firstly, it is not clear what the authors actually show with the steering experiments. In section 3.4, they state they want to test whether "the pre-CoT probes are merely correlated with the final answer, and do not themselves represent the final answer or causally influence it”. In Section 4.5 they state that "The results of the steering experiments are consistent with an interpretation of the pre-CoT probe as a causal representation of the pre- committed answer”. The preceeding Section 4.4 however does not make clear how the results actually imply this interpretation.
>
> We agree that our manuscript was not sufficiently clear about what we show with the steering experiments, and that the statement in Section 4.5, while not incorrect, does not follow transparently from the preceding results and therefore adds confusion. In particular, we did not clearly separate (1) empirical findings from (2) our proposed feature-level interpretation, nor did we clearly distinguish the strength of evidence we have for each.
>
> Concretely, the probe and steering experiments establish two empirical facts: (1) there is a direction in the pre-CoT residual stream that linearly decodes to the final answer, and (2) perturbing the model along this direction has a causal effect on the model’s eventual answer. A natural interpretation of this direction (and the one that motivated this work) is that it represents the model’s pre-committed answer. However, the feature-level claim “the probe direction is a representation of the pre-committed answer” is strictly stronger than the empirical hypothesis “the probe direction causally influences the final answer,” because the latter is necessary for the former. Our results directly confirm the empirical hypothesis and only offer suggestive evidence in support of the latter.
>
> When we wrote that our results are “consistent with an interpretation of the pre-CoT probe as a causal representation of the pre-committed answer,” our intent was to emphasize that a negative steering result (i.e., failing to change the answer when steering along the probe direction) would have counted as evidence against this interpretation. However, we recognize that the way this sentence was positioned did not make that logic clear.
>
> We see this confusion as a consequence of not having a dedicated Discussion section in the original draft. Empirical results and interpretive claims were interwoven in Section 4, which blurred the line between what we directly demonstrate about the probe directions and how we think those findings should be understood with respect to semantic interpretation. In the revision, we add a Discussion section and restrict the Results section to empirical findings only, while using the Discussion to focus on whether and how those findings support the “pre-committed answer” interpretation. We also revise the introduction to state our contributions explicitly as hypotheses, and we ensure that each results subsection is aligned with a specific hypothesis and presents evidence for that hypothesis.

---

> ### Author Response · Authors · 2025-12-04
> **Response to Reviewer aHJG (2/5)**
>
> >The way I understand it, the probe learns the directional difference of samples predictied “yes” versus samples predicted “no”. If we add/subtract this direction across all positions of the CoT chain, getting the answer to flip from “yes” to “no” and vice versa seems unsurprising.
>
> The reviewer’s understanding is correct in the sense that the probe learns a direction that separates pre-CoT activations for “yes” versus “no” predictions. However, we want to emphasize two points that we may have been unclear in the original draft.
>
> First, it is non-trivial that the model linearly separates the two answer classes *before* the CoT rather than only during it. Ultimately, at the final answer step, it must be the case that the model has linearly separated “yes” predictions from “no” predictions internally. That the latent representations of these two classes are linearly separable prior to CoT indicates a unique feature of how the model processes these examples.
>
> Second, it is also non-trivial that the learned direction is *instrumental* in generating the final answer, rather than merely reflecting some spurious correlational feature of the dataset. We can illustrate this by considering a hypothetical dataset, where the two classes are highly separable pre-CoT, but the learned directional difference is not instrumental in generating the final answer.
>
> For example, consider a dataset where all “yes” examples are written in Korean and all “no” examples are written in English. Suppose the task is hard enough that the model still requires CoT to answer correctly and that the model encodes language identity in its pre-CoT activations (a reasonable assumption of a sufficiently capable GPT-style language model). In this setting, the probe’s “yes–no” direction could simply correspond to a “Korean vs. English” direction, rather than a direction that truly represents the model’s final answer. However, steering along this direction would not reliably flip the answer from “yes” to “no” (or vice versa) in a targeted way, because semantically it would primarily influence the model’s belief about the text origin language, rather than the correct answer. Our steering results show that, in our setting, the probe direction is not behaving like such a spurious feature: changing activations along this direction reliably shifts the answer distribution.
>
> >For the causal claim “the pre-CoT activation along the probe direction  is causing the final model answer” (i.e. post-hoc commitment) it should be shown that “in the absence of the probe direction v* from the final pre-CoT position, the model flips it’s prediction”, i.e. proving causality by showing the counterfactual to reliably result in the opposite behaviour.
> >
> >To create this counterfactual, one could steer as , i.e. eliminate the linear component of "x_0" should along the “yes direction"  (and vice versa  for the “no direction”).
>
> Regarding the causal claim, our experiment and the reviewer's proposed test target two different notions of causality. In our setup, the steering coefficient $\alpha$ along the probe direction $v^\*$ is the causal variable and the model's final answer $Y$ is the outcome. We compare the baseline $\mathrm{do}(\\alpha = 0)$ (no steering) to interventions $\mathrm{do}(\alpha \neq 0)$ (steering along $v^\*$), and empirically observe that for many inputs
> $$
> P\\big(Y \\mid \\mathrm{do}(\\alpha = \\alpha_1)\\big) \\neq P\\big(Y \\mid \\mathrm{do}(\\alpha = 0)\\big)
> $$
> for sufficiently large $\\alpha_1$. This shows that manipulating $\\alpha$ is **sufficient** to change the answer distribution: changing only this variable induces a systematic change in $Y$. This is the sense in which we claim that the probe direction "causally influences" the model's final answer, and we further show that this influence is stronger than for equally large perturbations in orthogonal directions.
>
> By contrast, the reviewer's suggested experiment—ablating the projection onto $v^*$ at the final pre-CoT position and requiring the model to flip its prediction—tests a much stronger **necessity** claim. Informally, it asks us to show that "no $v^\*$ implies no effect," whose contrapositive is that observing the effect implies the presence of $v^\*$, i.e., $v^\*$ is necessary for the answer.
>
> In a system with redundant representations or multiple pathways to the same output, this necessity criterion can easily fail even if $v^\*$ genuinely participates in the mechanism that produces the answer. For this reason, we focus our empirical analysis on the weaker and more realistic sufficiency-style claim—demonstrating that steering along $v^\*$ can causally influence the answer—rather than on showing that $v^\*$ is strictly necessary for the model's behavior.
>
> (contd.)

---

> ### Author Response · Authors · 2025-12-04
> **Response to Reviewer aHJG (3/5)**
>
> That said, the necessity claim is still of interest. Actually, we implemented the suggested experiment in a preliminary exploration phase, but the results were underwhelming. Here, we present results from running the experiment again on 100 random examples for each model–dataset pair (not restricting to originally correct examples). We also include the below results in Appendix H.
>
> We report the frequency with which the model changes its answer. “Steering” denotes the rate of answer change under the ablating steering intervention, and “Baseline” denotes the rate of answer change when doing no intervention, i.e. the answer change rate due to sample stochasticity. “Effect” denotes the difference in answer change rate between Steering and Baseline. We include standard errors for each of the effect estimations. In cases where there was a parse failure for either the Steering generation or Baseline generation, we removed the example from the test results, resulting in N < 100 in many cases.
>
> In the majority of cases we do not see an effect size larger than random, though in a few cases the effect appears substantial, suggesting that the identified feature may sometimes satisfy this stronger necessity criterion.
>
> | Model         | Dataset              | AUC    | N   | Baseline    | Steering    | Effect  |
> |--------------|----------------------|--------|-----|-------------|-------------|---------|
> | Gemma 2 2B   | Anachronisms         | 0.9970 | 93  | 16.1±3.8%   | 16.1±3.8%   | +0.0%   |
> | Gemma 2 2B   | Logical Deduction    | 0.6884 | 98  | 30.6±4.7%   | 30.6±4.7%   | +0.0%   |
> | Gemma 2 2B   | Social Chemistry     | 0.9956 | 99  | 7.1±2.6%    | 7.1±2.6%    | +0.0%   |
> | Gemma 2 2B   | Sports Understanding | 0.9235 | 100 | 20.0±4.0%   | 19.0±3.9%   | -1.0%   |
> | Gemma 2 9B   | Anachronisms         | 0.9986 | 100 | 4.0±2.0%    | 4.0±2.0%    | +0.0%   |
> | Gemma 2 9B   | Logical Deduction    | 0.8778 | 100 | 3.0±1.7%    | 16.0±3.7%   | +13.0%  |
> | Gemma 2 9B   | Social Chemistry     | 0.9963 | 98  | 2.0±1.4%    | 1.0±1.0%    | -1.0%   |
> | Gemma 2 9B   | Sports Understanding | 0.9562 | 100 | 1.0±1.0%    | 2.0±1.4%    | +1.0%   |
> | Qwen 2.5 1.5B| Anachronisms         | 0.9882 | 86  | 14.0±3.7%   | 58.1±5.3%   | +44.2%  |
> | Qwen 2.5 1.5B| Logical Deduction    | 0.7070 | 93  | 32.3±4.8%   | 41.9±5.1%   | +9.7%   |
> | Qwen 2.5 1.5B| Social Chemistry     | 0.9931 | 87  | 8.0±2.9%    | 11.5±3.4%   | +3.4%   |
> | Qwen 2.5 1.5B| Sports Understanding | 0.8084 | 87  | 20.7±4.3%   | 26.4±4.7%   | +5.7%   |
> | Qwen 2.5 3B  | Anachronisms         | 0.9957 | 99  | 7.1±2.6%    | 15.2±3.6%   | +8.1%   |
> | Qwen 2.5 3B  | Logical Deduction    | 0.6895 | 100 | 13.0±3.4%   | 8.0±2.7%    | -5.0%   |
> | Qwen 2.5 3B  | Social Chemistry     | 0.9976 | 98  | 6.1±2.4%    | 5.1±2.2%    | -1.0%   |
> | Qwen 2.5 3B  | Sports Understanding | 0.9031 | 100 | 19.0±3.9%   | 14.0±3.5%   | -5.0%   |
> | Qwen 2.5 7B  | Anachronisms         | 0.9998 | 100 | 9.0±2.9%    | 8.0±2.7%    | -1.0%   |
> | Qwen 2.5 7B  | Logical Deduction    | 0.7777 | 99  | 6.1±2.4%    | 7.1±2.6%    | +1.0%   |
> | Qwen 2.5 7B  | Social Chemistry     | 0.9977 | 100 | 3.0±1.7%    | 0.0±0.0%    | -3.0%   |
> | Qwen 2.5 7B  | Sports Understanding | 0.9611 | 100 | 3.0±1.7%    | 1.0±1.0%    | -2.0%   |
>
> We understand that our original wording may have suggested a stronger notion of causality than we intended. In the revision, we clarify that our claim is that the probe direction *causally influences* the final answer, rather than that it is strictly necessary for computing the answer.

---

> > ### Author Response · Authors · 2025-12-04
> > **Response to Reviewer aHJG (4/5)**
> >
> > >(W3) Besides, the predictions seem to flip already a lot at random without any steering (alpha=0 in Figure 4), the orthogonal baseline results in even more flips, and the most reliable steering success usually happens at alpha levels with high parse failures. ... The high level of hallucination there further calls into question how general classification of steered CoT behavior really is.
> >
> > The frequency of flips without steering at alpha=0 rarely exceeds 20%. These frequencies are highest for smaller models, which is expected, as these models perform worse on the tasks, and accordingly we expect them to have greater uncertainty over the final answer, leading to greater answer variability on repeated trials. For the larger models (Gemma 2 9B and Qwen 1.5 7B) the answer flip rate at alpha=0 is very nearly 0 and quite negligible across tasks. We do not believe the frequency of flips without steering is a concern.
> >
> > While the effect of the orthogonal baseline is high in some cases, the relevant metric is the difference in flip rate between the steering intervention and the orthogonal baseline. This metric is positive across all model-dataset pairs in Figure 4. We also note that the orthogonal baseline is particularly low for the larger models Gemma 2 9B and Qwen 1.5 7B, providing confidence that the effect is not simply attributed to random intervention.
> >
> > That the most effective steering occurs when parse failures begin to emerge is an inevitable consequence of two expected phenomena: (1) that the effectiveness of steering grows with the steering coefficient and (2) at sufficiently large steering coefficients the model’s ability to follow instructions and reasoning generally will degenerate. However, in most cases, we observe the steering effect *before* parse failures increase. We acknowledge that the steering coefficient window for effectively controlling generations without ruining general reasoning ability is often narrow, limiting the practical usefulness of our method.
> >
> > Hallucination rates do increase with the steering coefficient, and some of this is clearly due to general reasoning collapse at large interventions. However, hallucinations are already more frequent than parse failures or orthogonal flip rates at small coefficients. This suggests that “hallucination” in our scheme does not just capture incoherent, collapsed reasoning.
> >
> > In particular, failed confabulation is also labeled as hallucination: when the model tries to justify the steered answer but produces false or non-entailing premises, the CoT is marked hallucinated even if it is otherwise coherent. Thus hallucination is an expected failure mode of confabulatory reasoning, not only of generic collapse. For this reason, we do not view hallucination rates alone as strong evidence against a genuine steering effect.
> >
> > > Section 4.2 references Appendix B but then proceeds with the same text as in the Appendix. Would it make sense to leave out this refence and just reference Appendix Figure 6?
> >
> > We have made this change.
> >
> > >Some citations seem to be formatted incorrectly ...
> >
> > We have made these fixes.
> >
> > >Is it possible that the models studied have already seen the evaluation data during training thus making successful post-hoc prediction more likely through data leakage?
> >
> > It is possible that the models studied may have seen the evaluation data during training, but we do not necessarily consider this a weakness for our experiments, as we wanted to find datasets where models had strong priors/biases about the answer. However, this consideration does incidence how these results might generalize "in the wild."
> >
> > >The link to the limitations in §5.2 does not point to the reported discussion of the probe caveats. Those appear in §5.1 (but probably should be in §5.2 as they also read like limitations).
> >
> > We have removed this link as well as the reference to the discussion of probe caveats. We have added a dedicated discussion section, which introduces the topics of probe interpretation and limitations itself, and we no longer find it necessary to introduce these ideas earlier in the text.
> >
> > >The Related Work section could use a clearer differentiation to the work presented in the paper. What are the gaps/limitations addressed through this work compared to existing works?
> >
> > In the related work we now distinguish between prompt-level studies of CoT unfaithfulness. We now make it more clear that our contribution is a mechanistic one.
> >
> > > Antropomorphisations like "To determine if the model is thinking about the final answer” seem unscientific. Personally, I would use a more exact wording like “if the final answer is represented”.
> >
> > We agree, and have modified this phrasing accordingly.
> >
> > >Why are some values for alpha=0 missing in Figure 5?
> >
> > We exclude alpha values for which there are fewer than 20 incorrect generations to classify. This detail was previously omitted, but is now included in the manuscript.

---

> > > ### Author Response · Authors · 2025-12-04
> > > **Response to Reviewer aHJG (5/5)**
> > >
> > > >Could you please clarify the hypothesis tested through the probing experiments and how the results imply this hypothesis exactly? Optimally, a mathematical definition of the variables thought to have a causal relation and how you test it.
> > >
> > > We want to clarify that the probe experiments alone (Section 4.3) do not test for causality, but merely if the final answer is linearly decodable from pre-CoT activations. After revisions, we state this hypothesis in the Introduction as H1.
> > >
> > > The steering experiments in Section 4.4 test the causality hypothesis H2, which we state as “Steering activations along this direction shifts the model’s answer far more than equally large orthogonal perturbations” in the introduction.
> > >
> > > We hope that the discussion of causality in response to weakness W2 is sufficiently formal to clarify what we mean by “causality”.
> > >
> > > >Would it make sense to look into further settings/hyperparameters for the steering to obtain results with a better ratio of successfully steered samples? In the light of the questions studied, focussing on one, optimal setting for alpha seems like a good idea. Right now, the results of 4.4 and 4.5 contain ablations for different alpha values of which most result in high parse failures or low steering success, making it hard to interpret them.
> > >
> > > We agree that identifying the optimal steering settings is important for the practical applications of this method. Specifically, the range of steering coefficients for which it is possible to reliably produce the desired effect (changing the model’s answer) without entirely degrading reasoning and instruction following ability is often small. However, to the best of our knowledge, it is difficult to a priori know which range of coefficients is most likely to be successful. We view our work as primarily about a study of interpretability, and as such leave research on improvements in control methods to other work.

---

### Official Review · Reviewer_bnMT · 2025-10-31

**Soundness:** 3
**Presentation:** 2
**Contribution:** 3
**Rating:** 4
**Confidence:** 3

**Summary:**

The paper investigates whether CoT faithfully reflects the model reasoning capabilities. This work shows that models pre-commit to an answer before generating the CoT and train simple contrastive linear probes on the activation of the model and perform activation steering. Through steering the model flips its answer in more than 50% of the original examples, showing that model actually uses this representation.
The paper further studies how the intervention through activation steering causes the model to change its answer.

**Strengths:**

1. The high pre-COT probe performance shows an interesting observation of the existence of unfaithful COT.
2. I found the CoT classification analysis interesting in how the model changes its CoT, which leads to a change in the answer.

**Weaknesses:**

1. Even though authors show the existence of the final answer pre-CoT, they use simple classification tasks. I am doubtful that this might not hold up for tougher datasets. Does the author show the performance difference between CoT and Non-CoT answer?
2. The paper only studies normal models; however, results might be different in reasoning models that undergo multiple rounds of self-verification and backtracking to reach an answer.
3. While testing for CoT sensitivity, how is the swapping of the CoT taking place? Specifically, how is CoT modified to introduce mistakes?
4. In Figure 4, why is there a difference in trends across models, for example in task sports understanding gemma 2 2B shows how count of flips even in large steering coefficient, however this does not hold up for qwen2.5-1.5B.
5. What is the motivation to pick these 4 tasks?

**Questions:**

NA

---

> ### Author Response · Authors · 2025-12-03
> **Response to Reviewer bnMT**
>
> We thank the reviewer for the thoughtful comments and invitations for clarification.
>
> >Even though authors show the existence of the final answer pre-CoT, they use simple classification tasks. I am doubtful that this might not hold up for tougher datasets. Does the author show the performance difference between CoT and Non-CoT answer?
>
> Per the pooled response, we’ve now added non-CoT accuracies to Table 2. Crucially, the Logical Deduction task benefits the most from CoT, indicating that models rely on the CoT to compute answers for this task more than others. Accordingly, probes on Logical Deduction have much lower test AUC than probes on other tasks. However, we do not view this as a weakness of our method but rather evidence that it is working correctly. If we assume that the final answer is not strongly encoded pre-CoT for Logical Deduction, then high scoring probes would be false positives, likely capturing dataset artifacts or other spurious correlations. That the probes have much less discriminatory power and steering is less effective relative to baseline for the Logical Deduction task increases confidence that successful probes and steering are properly capturing a representation of the pre-committed answer.
>
> >While testing for CoT sensitivity, how is the swapping of the CoT taking place? Specifically, how is CoT modified to introduce mistakes?
>
> For each original generation, we use an LLM (GPT-5) to modify the original CoT to introduce a mistake that implies the opposite answer. The LLM is instructed to modify the original CoT minimally—for example, we encourage adding  negations or swapping words to change the meaning of the CoT without substantially changing the text. Ideally, we would like the incorrect CoT to resemble the original CoT to the greatest extent possible, to minimize the probability that the CoT is perceived as “foreign”. We have added a new Appendix B detailing the CoT sensitivity methods.
>
> >In Figure 4, why is there a difference in trends across models, for example in task sports understanding gemma 2 2B shows how count of flips even in large steering coefficient, however this does not hold up for qwen2.5-1.5B.
>
> We agree that Figure 4 shows heterogeneous trends across model–dataset pairs. In general, we do not expect the steering curves to look uniform across all models and tasks. There are several moving parts: we select different probe layers for each model–dataset pair; some probes will, for idiosyncratic reasons, generalize better or align more cleanly with the targeted semantic feature than others; and different models can respond to strong interventions in activation space in qualitatively different and sometimes unpredictable ways. Moreover, subtle semantic differences between probe directions can be amplified at higher steering coefficients, which can accentuate these model-specific behaviors.
>
> Our primary claim about Figure 4 is not that all curves should match in shape, but that the *overall* effectiveness of steering tends to grow with the magnitude of the steering coefficient for each model–dataset pair. The global trend that stronger steering generally induces more flips in the target direction is the main phenomenon we intend to highlight, and it holds across the models we study despite their idiosyncratic differences.
>
> >What is the motivation to pick these 4 tasks?
>
> First, we wanted settings where chain-of-thought (CoT) is differentially useful: The accuracies we now report in Table 2 show that CoT clearly helps on some tasks (e.g., Logical Deduction for larger models), has little effect on others, and can even hurt performance in some cases.
>
> Second, we sought diversity in domain and reasoning style: the tasks span temporal consistency and commonsense (Anachronisms), formal reasoning (Logical Deduction), social and normative judgments (Social Chemistry), and applied factual reasoning (Sports Understanding), which lets us study post-hoc reasoning mechanisms across qualitatively different types of prompts.
>
> Third, we required tasks that could be operationalized as binary classification problems, so that we could define a simple difference-of-means probe and interpret positive vs. negative steering in a straightforward way.

---

### Official Review · Reviewer_Hauf · 2025-11-02

**Soundness:** 2
**Presentation:** 2
**Contribution:** 2
**Rating:** 4
**Confidence:** 4

**Summary:**

This paper investigates the faithfulness of chain-of-thought (CoT) reasoning in large language models—whether the model’s verbalized reasoning truly reflects its decision-making process. While CoT improves interpretability and performance, prior work shows models often generate misleading or post-hoc rationales. The authors study post-hoc reasoning, where answers are determined before reasoning begins. Through experiments, they identify two phenomena: (1) Pre-CoT probes reveal that final answers are linearly decodable from activations before reasoning starts, implying answer pre-commitment; and (2) Answer steering shows that manipulating these activations can flip answers, often producing incoherent or non-entailing CoTs, evidencing unfaithful reasoning.

**Strengths:**

The paper addresses an important question regarding the faithfulness of chain-of-thought (CoT) reasoning. Its use of pre-CoT probes and answer steering as tools to examine post-hoc reasoning offers a creative and insightful approach that goes beyond merely identifying unfaithful CoT. The answer steering experiments, in particular, provide a meaningful and valuable extension to current studies on CoT faithfulness. The work could make a more substantial contribution to the field with a deeper comparative analysis encompassing both LLMs and LRMs.

**Weaknesses:**

1. The paper’s writing requires significant improvement. The related work section is incomplete, missing key prior studies on chain-of-thought (CoT) faithfulness (as mentioned in questions). The presentation of results and conclusions lacks logical flow, making it difficult to understand the causal link between hypotheses, methods, and findings.
2. The experimental design is shallow relative to prior work. The “model knows the answer before CoT” phenomenon has been well documented, and the presented “pre-CoT probe” and “answer steering” analyses do not substantially advance the empirical understanding beyond existing results.
3. The paper does not concretely connect its findings to practical principles or solutions that could guide future research to improve the reasoning ability of LLMs.

**Questions:**

LN076: What is the novel part of the work compared to existing related work?

LN086: Missing high relevant related work: "How Likely Do LLMs with CoT Mimic Human Reasoning?" which also reports the issue that LLMs may know the answer before the CoT and studies it empirically.

LN131: It is also very similar to the intervention used by the related work mentioned above.

LN229: These accuracies are before interventions. What is the point? How does it relate to your claim?

LN355: What is the unit for the y-axis?

LN350: Are these models working in think (long CoT) or non-think mode?

LN417: The Conclusion section actually contains discussions instead of any conclusion.

---

> ### Author Response · Authors · 2025-12-03
> **Response to Review Hauf (1/2)**
>
> We thank the reviewer for the careful review. We found feedback on the paper's writing and structure particularly helpful, and believe our revisions have led to a much stronger manuscript. Below we address several specific comments.
>
> ---
>
> >The paper’s writing requires significant improvement. The related work section is incomplete, missing key prior studies on chain-of-thought (CoT) faithfulness (as mentioned in questions). The presentation of results and conclusions lacks logical flow, making it difficult to understand the causal link between hypotheses, methods, and findings.
>
> We agree that some relevant prior art was missing from our Related Work section. We have added the suggested missing prior art to the Related Work section.
>
> To address the concerns about logical flow and the causal link between hypotheses, methods, and findings, we have also restructured the paper. As described in our pooled response on paper organization, we now explicitly state one empirical premise (P0) and three hypotheses (H1–H3) in the introduction, and we tie each subsection of Section 3 (Methods) and Section 4 (Results) to the specific hypothesis it tests.
>
> >The experimental design is shallow relative to prior work. The “model knows the answer before CoT” phenomenon has been well documented, and the presented “pre-CoT probe” and “answer steering” analyses do not substantially advance the empirical understanding beyond existing results.
>
> We agree that the existence of pre-CoT answer commitment can be demonstrated with prompt-level interventions, and we do not claim novelty for that phenomenon itself. Our goal is to move from a behavioral observation (“the model must have decided somewhere before the CoT”) to a more mechanistic statement about how that decision is represented and used.
>
> Concretely, we ask: is the final answer linearly decodable from pre-CoT activations, and does that same direction play a causal role in determining the answer? The pre-CoT probe experiments directly address the first question, while the steering and CoT-classification experiments are designed to interpret this feature.
>
> Identifying a single, linearly decodable feature that both predicts and controls the final answer across models and tasks, and characterizing how CoT changes when that feature is perturbed, is the empirical advance we aim to provide over prior behavioral demonstrations. While this result is simple, we do not believe it to be insignificant. Linear representation both rules out competing hypotheses (e.g., that the answer is represented non-linearly prior to CoT), and suggests opportunities to monitor for post-hoc reasoning (e.g., probes) and control post-hoc reasoning (e.g., steering).
>
> >The paper does not concretely connect its findings to practical principles or solutions that could guide future research to improve the reasoning ability of LLMs.
>
> Our object in this paper is to understand unfaithful CoT; this is conceptually distinct from improving reasoning ability in the sense of task accuracy, and we view our work as primarily diagnostic rather than algorithmic. Indeed, sometimes improving faithfulness may be at odds with improving reasoning ability. We strongly suspect, for instance, that post-hoc reasoning is learned precisely because it is useful toward arriving at the correct response.
>
> That being said, there may be instances where improving faithfulness (specifically, via mitigating post-hoc reasoning) results in improved reasoning. For example, models may learn miscalibrated priors, undesirable biases, or incorrect facts that are present in training data, and encode these beliefs before CoT. In turn, this could result in incorrect responses that would not have been produced had the model engaged the CoT, and not reasoned about the answer post-hoc. Controlling the pre-CoT answer feature could plausibly induce the model to reason from a blank slate, forcing it to depend upon the CoT to generate its response.
>
> We agree that describing these instances would strengthen our paper, and describe the example from above more briefly in our suggestions for future work.

---

> > ### Author Response · Authors · 2025-12-04
> > **Response to Review Hauf (2/2)**
> >
> > >Missing high relevant related work: "How Likely Do LLMs with CoT Mimic Human Reasoning?" which also reports the issue that LLMs may know the answer before the CoT and studies it empirically.
> >
> > Thank you for pointing us to this paper. We have included it in our background section.
> >
> > >It is also very similar to the intervention used by the related work mentioned above.
> >
> > [Bao et al.](https://arxiv.org/abs/2402.16048) use prompt interventions to construct causal graphs of CoT where the nodes are the instruction prompt, the CoT, and the answer. They further classify these graphs according to the reasoning style they represent, and comment on the faithfulness and internal consistency of each reasoning style.
> >
> > Conceptually, this work is very similar to ours in that it explores the causal structure of reasoning by breaking it down into three components: pre-CoT instruction, CoT, and answer. However, this is the natural decomposition of CoT reasoning, and Bao et al. are not the first to study CoT through the lens of causal analysis. Our paper cites [nostalgebraist](https://www.lesswrong.com/posts/HQyWGE2BummDCc2Cx/the-case-for-cot-unfaithfulness-is-overstated) who invokes the same decomposition to discuss CoT unfaithfulness, and [Gao](https://www.lesswrong.com/posts/FX5JmftqL2j6K8dn4/shapley-value-attribution-in-chain-of-thought)who constructs causal networks by perturbing intermediate results in multi-step arithmetic problems.
> >
> > This is to say that while Bao et al. is highly relevant to our work and ought to be included in the background section, it does not introduce significant new concepts to the background, and the context in which we situate our work remains the same.
> >
> > Our methods differ from those used in Bao et al. in the same way they differ from those in Gao in that they involve activation-level intervention and analysis as opposed to prompt-level.
> >
> > >These accuracies are before interventions. What is the point? How does it relate to your claim?
> >
> > We now included accuracies without CoT beside the accuracies with CoT in Table 2, and in Section 3.2, we directly relate the accuracy reporting to the premise P0. Specifically, by demonstrating that on some tasks, several models do not appear to benefit from using CoT, we establish evidence that the model may be insensitive to CoT. Establishing a reasonable belief that the model may not use the CoT to compute its final answer provides the foundation for our exploration of how post-hoc reasoning occurs mechanistically.
> >
> > >What is the unit for the y-axis?
> >
> > The y-axis is the answer flip rate, the frequency with which the model changes its answer from the original. We have added this to the figure caption.
> >
> > >Are these models working in think (long CoT) or non-think mode?
> >
> > None of the models studied in the main results are reasoning models, and do not possess a "think" mode.
> >
> > >The Conclusion section actually contains discussions instead of any conclusion.
> >
> > We have restructured the paper so that there is now a proper discussion section, which contains the content that was originally included in the conclusion. The conclusion now restates the structure of the paper and summarizes findings in a conventional manner.

---

### Official Review · Reviewer_WttT · 2025-11-06

**Soundness:** 3
**Presentation:** 4
**Contribution:** 3
**Rating:** 6
**Confidence:** 4

**Summary:**

This paper studies the post-hoc reasoning phenomena in language models, where the language model uses its CoT to reason in favor of a pre-determined answer (instead of doing computations that lead to an answer). It shows that the final answer of model is linearly decodable from the pre-CoT activations inside the model. Moreover, it shows that steering those activations could result in the model changing its originally correct answer, hence concluding that the model actually uses the extracted features to determine its final answer. In these steering cases, the paper classifies CoT pathologies into confabulation (false premises supporting the flipped answer) and non-entailment (true premises that do not lead to the conclusion) cases, giving insight into how post-hoc reasoning is reflected in the CoT.

**Strengths:**

The paper studies an important problem in reasoning literature that has implications about faithfulness of language models. It is comprehensive in the study, by studying both prediction from representations, as well as their causal role in determining the final answer. Moreover, the paper gives insight into how the post-hoc reasoning manifests in the CoT by classifying the steered cases into two cases, making it useful for studying potentials and limitations of CoT monitoring methods.

**Weaknesses:**

1. In the steering experiments, the best layer for steering is chosen using the test dataset, which might cause selection bias (especially with the relatively small test dataset). Instead, a validation dataset should have been used to select the best layer and then report the results on test set.
2. The tasks are chosen such that CoT does not help much with accuracy, it would be nice to include a datasets where CoT actually improves performance. The current datasets are limited to binary classification.
3. The generations are limited to correct ones, while the post-hoc reasoning behavior could happen no matter the model’s pre-determined answer is correct or not.

**Questions:**

1. In the logical decuctoin experiments, the pre-CoT probe achieves less AUC, potentially because the final answer depends more on the CoT. However, the “Ellipses” intervention does not affect it much. How do you explain this?
2. Why have you limited the generations to correct ones?
3. Could you provide examples of the cases labeled as confabulation and non-entailment? Did you verify that the labeling is accurate?

---

> ### Author Response · Authors · 2025-12-03
> **Response to Reviewer WttT (1/2)**
>
> We appreciate this thoughtful review. We address several of the reviewer's weaknesses and questions below.
>
> ---
>
> >In the steering experiments, the best layer for steering is chosen using the test dataset, which might cause selection bias (especially with the relatively small test dataset). Instead, a validation dataset should have been used to select the best layer and then report the results on test set.
>
> This is a useful criticism, and we agree that, in principle, the steering layer should be selected on a validation set rather than the test set. Due to time and compute constraints, we are not able to re-run the full steering pipeline with newly computed probes for every configuration, but we directly address the selection-bias concern by re-training and re-evaluating probes on proper train–validation–test splits and comparing the results to those used in the paper.
> For each model–dataset pair, we take the original 500 train + 500 test examples (1,000 total) and randomly split them into a train set ($n=400$), validation set ($n=400$), and test set ($n=200$). We train probes on the train set, select the layer according to validation AUC, and then evaluate the chosen probe on the held-out test set. We then compare the selected layer and test AUC against our original probe results.
>
> Under this procedure, we select the same probe layer for the majority of model–dataset pairs:
>
> | Metric                   | Value      |
> |--------------------------|------------|
> | Same layer selected      | 12 (54.5%) |
> | Different layer selected | 10 (45.5%) |
>
>
> When the selected layers differ, the discrepancy is small:
>
> | Layer Difference | Value |
> |------------------|-------|
> | Max              | 3     |
> | Median           | 0.0   |
> | Mean             | 0.77  |
>
>
> Finally, the resulting test AUCs differ very little:
>
> | Test AUC Difference (Δ) | Value    |
> |-------------------------|---------|
> | Max                     | +0.0994 |
> | Min                     | -0.0131 |
> | Median                  | 0.0000  |
> | Mean                    | +0.0105 |
>
> While our experimental method is imperfect, we believe these results indicate that any selection bias introduced by our original procedure is minor and unlikely to materially affect our conclusions.
>
> >The tasks are chosen such that CoT does not help much with accuracy, it would be nice to include a datasets where CoT actually improves performance. The current datasets are limited to binary classification.
>
> We have updated Table 2 in Section 4.1 to report accuracy with and without chain-of-thought (CoT) for all model–dataset pairs. This makes clear that the usefulness of CoT varies across both models and tasks. For instance, Logical Deduction shows the largest gains from CoT (for sufficiently large models), whereas CoT is unhelpful or even slightly harmful on Social Chemistry.
>
> More broadly, we observe that the discriminative power of the pre-CoT probes is anti-correlated with the usefulness of CoT: when CoT does not help much, the final answer is easier to decode from pre-CoT activations. Logical Deduction provides a counterpoint—here, pre-CoT probes are comparatively weak exactly when the model relies on CoT to compute the answer.
>
> We agree that restricting our study to binary classification tasks is a limitation. This choice was intentional: binary classification substantially simplified the experimental design and analysis. In particular, it allowed us to use a single difference-of-means probe per task, and made positive vs. negative steering coefficients straightforward to interpret. We encourage future work that extends our experiments beyond binary classification tasks!

---

> > ### Author Response · Authors · 2025-12-04
> > **Response to Reviewer WttT (2/2)**
> >
> > >The generations are limited to correct ones, while the post-hoc reasoning behavior could happen no matter the model’s pre-determined answer is correct or not.
> >
> > It is true that post-hoc reasoning is not limited to instances where the model’s answer is correct. However, there are a few reasons why we intentionally focus on steering originally correct examples toward the incorrect answer.
> >
> > First, we strongly suspect that changing correct answers to be incorrect is a more difficult task than changing incorrect answers to be correct. This is because we expect the median confidence of the model on correct responses to be higher than on incorrect responses, under the assumption that the model has better-than-random calibration on our datasets. As such, we believe that steering correct answers toward incorrect ones offers a stronger signal about post-hoc reasoning than steering over all model answers.
> >
> > Second, we are particularly interested in how chain-of-thought (CoT) unfaithfulness can lead to harmful or deceptive behaviors. Holding a strong prior over the final answer is typically benign when that belief is correct. In contrast, when the model is pushed toward an incorrect belief, it must either confabulate facts or engage in illogical reasoning in order to rationalize that belief. Focusing on examples where the model was originally correct therefore allows us to concentrate on these undesirable behaviors.
> >
> > Lastly, on its own, the CoT trace of a model that has been steered toward the correct answer offers less information about whether the model’s reasoning is unfaithful. In such cases, the model may simply state a correct argument, which does not by itself indicate whether the probe influences the model’s intermediate steps, its conclusion, both, or neither.
> >
> > >In the logical decuctoin experiments, the pre-CoT probe achieves less AUC, potentially because the final answer depends more on the CoT. However, the “Ellipses” intervention does not affect it much. How do you explain this?
> >
> > As mentioned in the pooled response, we have updated Table 2 to report the accuracy with and without CoT, confirming that the final answer depends more on the CoT in the Logical Deduction examples.
> >
> > We have also re-run the CoT sensitivity experiments to include Qwen 2.5 3B and use an improved “Mistakes” incorrect CoT generation workflow.
> >
> > It is true that the ellipses intervention causes the model to flip its answer less than we might expect on the Logical Deduction task, given that the CoT is responsible for approximately a 24% increase in accuracy for Gemma 2 9B, 11% for Qwen 2.5 3B, and 10% for Qwen 2.5 7B. However, an important distinction between the “Ellipses” intervention and the “no-CoT” generations from Table 2 is that the four in-context demonstrations in the Ellipses prompt contain chain-of-thought, whereas the in-context demonstrations in the no-CoT demonstrations do not. We hypothesize that this makes the Ellipses task easier than the no-CoT task: Although in both cases the model is not allowed CoT during generation, the CoT demonstrations in Ellipses are more didactic than the no-CoT demonstrations.
> >
> > >Could you provide examples of the cases labeled as confabulation and non-entailment? Did you verify that the labeling is accurate?
> >
> > In lieu of a comparison with a human classifier, we verify the quality of the labeling by measuring the internal consistency of GPT-5-mini. In this process, we decided our classification prompt could be improved, so we made these improvements and re-ran the classification on all examples. Then, we chose a random subset of 200 generations to classify twice, and compared those classifications to the original classifications. We report the consistency findings in Appendix E.
> >
> > We find that GPT-5-mini is fairly consistent in labeling the two dimensions of the classification scheme: 92.0% for false premises and 89.5% for conclusion entailment. When these two dimensions are combined to form the final CoT classification according to Table 1, consistency is a bit lower at 81.5%.
> >
> > In Appendix E we also provide a random sample of six CoT classifications, with examples of confabulation, non-entailment, and hallucination.
> >
> > In addition to reporting these statistics, we note creating the classification prompt was a trial-and-error process. Though we did not rigorously grade prompt versions ourselves, the process did involve manual inspection of input-output pairs to ensure the classifications and their explanations were satisfactory.
> >
> > Ultimately, there is a lot of room for ambiguity in our classification regime, and some label inconsistency is expected and perhaps irreducible. However, the role of the classification results in our paper is to illustrate general trends (e.g., hallucination rates increasing over steering strength) and show non-negligible rates of confabulation and non-entailment. As such, we do not think minor classification inconsistency meaningfully detracts from our findings.

---

### Author Response · Authors · 2025-12-03
**Response to all reviewers**

1. **Task accuracy with and without CoT**

Several reviewers suggested comparing task accuracy with and without chain-of-thought (CoT), and raised concerns that our benchmarks might not span a range of CoT usefulness. In response, we now report both CoT and no-CoT accuracies in Table 2 of the revised manuscript, whereas the original submission only contained CoT accuracies.

The usefulness of CoT—measured as the difference between CoT and no-CoT accuracy—varies substantially across tasks and models. Logical Deduction shows the largest and most consistent gains from CoT, whereas CoT offers little or even negative benefit on Anachronisms. This pattern is consistent with our probe results: Logical Deduction has comparatively weak pre-CoT probes, suggesting that for this task the model often computes its final answer during the CoT rather than beforehand.

2. **No use of reasoning models**

Several reviewers noted the omission of reasoning models. Our original focus on non-reasoning, instruction-tuned models was intentional, for both practical and conceptual reasons. We have now included results for a large reasoning model (GPT-OSS 20B) in Appendix E, and these results reinforce our initial rationale.

First, preliminary experiments suggested that steering was substantially less effective in reasoning models, making it difficult to draw clear conclusions using the same methodology. The new GPT-OSS 20B results confirm this: probe AUCs are markedly lower across nearly all datasets (except Anachronisms), and steering produces negligible answer-flip rates relative to the orthogonal baseline.

Second, narrowing the main scope to non-reasoning models allows us to analyze post-hoc reasoning with much greater nuance. Reasoning models introduce additional design decisions, e.g., whether to probe before or after "thinking" tokens, how to steer CoT tokens, how to structure demonstrations, that would significantly expand the paper’s methodological surface area. Including these choices in the main body would dilute the interpretability of our results. For this reason, we keep the core analysis on non-reasoning models and place the limited LRM results in the appendix.

Third, our central motivation concerns optimization pressures that are most salient for non-reasoning models: their CoT is directly visible and subject to human feedback during RLHF, creating incentives for explanations to become persuasive or human-pleasing. For LRMs such as GPT-OSS 20B, our appendix findings are consistent with a different dynamic. We hypothesize that LRMs concentrate load-bearing computation inside the CoT itself, which aligns with the weak representation of pre-committed answer directions in pre-CoT activations and the ineffectiveness of steering, even in the one dataset (Anachronisms) where probe AUC is high.

Finally, while our results are not centered on reasoning models, we do not believe they are irrelevant for them. Post-hoc reasoning behaviors in non-reasoning models may be distilled into LRMs, and tools such as modifying prior beliefs or ablating memorized incorrect answers prior to CoT could plausibly translate. The GPT-OSS 20B appendix results help illustrate where these connections might hold and where LRMs diverge.

3. **Paper organization, hypotheses tested, and experiment interpretation**

We found this feedback particularly helpful and have substantially reorganized the paper in response.

- We now explicitly state one empirical premise (P0) and three hypotheses (H1–H3) in the introduction. P0 summarizes prior evidence that models can “know” the answer before CoT and that CoT is differentially useful across tasks, which we verify on our datasets. H1 posits that the model’s final answer is encoded and linearly decodable from pre-CoT residual activations. H2 posits that this direction has a specific causal influence on the final answer (steering along it changes the answer more than orthogonal perturbations). H3 concerns the qualitative failure modes that arise when we steer toward the incorrect answer (confabulation and non-entailment in the generated CoT).
- Each subsection of Section 3 (Methods) is now directly linked to the hypothesis that its experiment is designed to test.
- Each subsection of Section 4 (Results) is explicitly connected to the corresponding hypothesis interpretation, so that the logical flow from premise → hypothesis → method → result is clearer.
- We added a dedicated Discussion section and moved the detailed interpretation of features and broader implications out of the Results section into this new section, to better separate empirical findings from higher-level interpretation.

Including P0 alongside H1–H3 clarifies the boundary between prior literature and our contribution. We do not claim the discovery of post-hoc reasoning as novel; instead, our work focuses on testing hypotheses about the *mechanisms* underlying post-hoc reasoning.

---

### Meta-Review · Area_Chair_mKnQ · 2026-01-05

**Summary:**

The paper considers the role of post-hoc reasoning CoTs, wherein answers are pre-determined and the reasoning trace is retrofit against this. It constructs experiments to test the hypothesis that there is strong information in pre-CoT tokens that influence the final answer, which represents a causal link that results in predictable pathologies when steered away from the right answer.

Reviewers appreciated the problem and detailed empirical study, but also had a number of concerns:
- **Restriction to non-reasoning models**. Multiple reviewers noted that the results for reasoning models (which were studied in related work) may be significantly different. As reasoning models are trained to optimize the CoT, their omission was seen as surprising.
- **Restriction to binary tasks where CoT has limited value**. Multiple reviewers noted that the value of CoT in the tasks chosen was not clear, and/or appeared to be limited. The restriction to binary classification tasks was also seen as a concern.
- **Steering experiments setup**. One reviewer expressed concern at the setup for the steering experiments (a key contribution), noting that it was unclear how the presented results were in concordance with the claims made.
- **Clarity of exposition**. Multiple reviewers expressed confusion at how some of the claims were presented.
- **Discussion and novelty over prior works**. One reviewer noted that there is a missing relevant work, _How Likely Do LLMs with CoT Mimic Human Reasoning_, which has very similar methodologies.

**Reviewer Concerns:**

- **Restriction to non-reasoning models**. The authors presented a detailed motivation for the focus on non-reasoning models, owing to the greater ease of steerability, ability to conduct more nuanced analyses over fewer variables, and their direct optimization pressure on the CoT.
  - *Partially addressed*. The authors' arguments are not without merit. However, as noted, prior work in the area has used reasoning models. Given that such models are the dominant surface where CoTs are used, reviewers' concern on this point is understandable.
  - From our reading, another minor remark is that Gemma-2 should perhaps be replaced with Gemma-3.
- **Restriction to binary tasks where CoT has limited value**. The authors provided detail on CoT versus non-CoT accuracies across tasks, showing that in certain cases there is a big gap. Further, in such cases, the probes have limited discriminatory power, from which the authors gain confidence that the method does not pick upon spurious correlations. The authors acknowledged the focus on binary classification tasks to be a limitation that could be of interest to explore in future work.
  - *Mostly addressed*. The response on this point seems satisfactory.
- **Steering experiments setup**. The authors presented a lengthy clarification and discussion of the results, as well as updates to the paper.
  - *Partially addressed*. The new separation of the steering results and their interpretation are reasonably clear. However, as noted by the authors in Section 5.1, there are limits to what can be concluded in regards to hypothesis H3.
- **Clarity of exposition**. The authors presented a series of updates, including a separation of the experimental results and their discussion.
  - *Partially addressed*. The updates have helped the paper, although they seem to have involved a few non-trivial structural changes to the original manuscript. From our reading, there is still some scope for greater clarity: for example, the outline of the overall narrative in the Conclusion is actually much clearer than the preceding sections. This is perhaps owing to the hypotheses H1 -- H3 not being explicitly mentioned in Section 4 and 5, and Section 5 mixing a discussion of results with limitations of the same (which is commendable, but we posit could be done in a clearer way).
- **Discussion and novelty over prior works**. The authors acknowledged that their related work section was incomplete. However, they argued that the methodology of referenced works is still different to what they consider, particularly in terms of going from prompt- to activation-level intervention.
  - *Partially addressed*. A more precise technical explanation of how the interventions in this paper differ from that of prior work would have been useful.

**Reviewer Scores:**

- **bnMT**: the reviewers' primary concerns were around the lack of reasoning models, and the tasks considered being in a certain sense "easy". As these were only addressed partly, we tend to think the reviewer's score would have remained at 4.
- **WttT**: The review was generally positive, and the questions were reasonably addressed. Thus, we could imagine that the reviewer's score would have increased to 8.
- **Hauf**: the reviewers' primary concerns were around clarity and exposition, relation to prior work, and the practical take-aways from the paper. We tend to think the reviewer's score would have remained at 4.
- **aHJG**: the reviewers' primary concerns were around the lack of reasoning models, and questions around the steerability analysis. The response did somewhat address the latter point. We tend to think it would have remained at 4 given the number of detailed concerns, but we could imagine the reviewer possibly increasing their score to 6.

---

### Decision · Program_Chairs · 2026-01-26

Reject